# [99mTc]Tc-Sestamibi Bioaccumulation Can Induce Apoptosis in Breast Cancer Cells: Molecular and Clinical Perspectives

**Nicoletta Urbano** [1,†], **Manuel Scimeca** [2,3,4,†] [iD], **Rita Bonfiglio** [5], **Alessandro Mauriello** [2,5] [iD], **Elena Bonanno** [2,6] and **Orazio Schillaci** [7,8,*]

1    Nuclear Medicine Unit, Department of Oncohaematology, Policlinico "Tor Vergata", 00133 Rome, Italy; n.urbano@virgilio.it
2    Department of Experimental Medicine, University of Rome "Tor Vergata", Via Montpellier 1, 00133 Rome, Italy; manuel.scimeca@uniroma2.it (M.S.); alessandro.mauriello@uniroma2.it (A.M.); elena.bonanno@uniroma2.it (E.B.)
3    San Raffaele University, Via di Val Cannuta 247, 00166 Rome, Italy
4    Saint Camillus International University of Health Sciences, Via di Sant'Alessandro, 8, 00131 Rome, Italy
5    Tor Vergata Oncoscience Research (TOR), University of Rome "Tor Vergata", 00133 Rome, Italy; rita.bonfiglio@uniroma2.it
6    Diagnostica Medica & 'Villa dei Platani', Neuromed Group, 83100 Avellino, Italy
7    IRCCS Neuromed, Via Atinense, 18, 8607 Pozzilli, Italy
8    Department of Biomedicine and Prevention, University of Rome "Tor Vergata", Via Montpellier 1, 00133 Rome, Italy
\*    Correspondence: orazio.schillaci@uniroma2.it
†    Nicoletta Urbano and Manuel Scimeca equally first author.

**Abstract:** The aim of this study was to investigate the possible role of [99mTc]Tc-Sestamibi in the regulation of cancer cell proliferation and apoptosis. To this end, the in vivo values of [99mTc]Tc-Sestamibi uptake have been associated with the in-situ expression of both Ki67 and caspase-3. For in vitro investigations, BT-474 cells were incubated with three different concentrations of [99mTc]Tc-Sestamibi: 10 µg/mL, 1 µg/mL, and 0.1 µg/mL. Expression of caspase-3 and Ki67, as well as the ultrastructure of cancer cells, was evaluated at T0 and after 24, 48, 72, and 120 h after [99mTc]Tc-Sestamibi incubation. Ex vivo data strengthened the known association between sestamibi uptake and Ki67 expression. Linear regression analysis showed a significant association between sestamibi uptake and the number of apoptotic cells evaluated as caspase-3-positive breast cancer cells. As concerning the in vitro data, a significant decrease of the proliferation index was observed in breast cancer cells incubated with a high concentration of [99mTc]Tc-Sestamibi (10 µg/mL). Amazingly, a significant increase in caspase-3-positive cells in cultures incubated with 10 µg/mL [99mTc]Tc-Sestamibi was observed. This study suggested the possible role of sestamibi in the regulation of pathophysiological processes involved in breast cancer.

**Keywords:** 99mTc-sestamibi; breast-specific γ imaging; breast cancer; apoptosis; theragnostic

## 1. Introduction

A recent report by the American College of Radiology released the clinical indications for the use of dedicated breast γ imaging, including breast-specific γ imaging (BSGI), in breast cancer patients not suitable for magnetic resonance imaging (MRI) analysis [1]. Indeed, BSGI with the cationic lipophilic agent [99mTc] (technetium 99-m) labeled sestamibi (99mTc-Sestamibi) is considered a reliable and useful medical device for the early preoperative detection of primary breast cancer [2]. BSGI with [99mTc]Tc-Sestamibi can also be considered complementary to mammography in women showing dense breasts, palpable abnormalities, and mammographically indeterminate breast lesions > 1 cm [3]. For all these reasons, the correct use of BSGI with [99mTc]Tc-Sestamibi in the breast units frequently allows for a reduction in the number of unnecessary biopsies, thus improving

the management of breast cancer patients. Moreover, it established the role of BSGI with [99mTc]Tc-Sestamibi in accurate stereotactic breast biopsy procedures. In fact, Collarino and colleagues [4] described a new clinical method for stereotactic breast biopsy guided by [99mTc]Tc-Sestamibi uptake that allows for the achievement of larger and suitable histopathologic samples, if compared with automated core needle biopsy methods [5,6].

Recently, Urbano et al. demonstrated that BSGI with [99mTc]Tc-Sestamibi can be used to detect breast cancer lesions characterized by the presence of microcalcifications and breast osteoblast-like cells [7]. Indeed, authors showed a significant association between [99mTc]Tc-Sestamibi uptake, the presence of breast microcalcifications made of hydroxyapatite, and the presence of cancer cells associated with the development of bone metastatic lesions breast osteoblast-like cells [8]. Thus, BSGI with [99mTc]Tc-Sestamibi could also be used for the early detection of breast cancer lesions with a high propensity to form bone metastasis [9].

In regard to [99mTc]Tc-Sestamibi uptake, two different biodistribution models based on its chemical-physical characteristics have been proposed [10–12]: (a) binding of [99mTc]-Tc-Sestamibi with 8–10-kDa cytoplasmic proteins, and (b) easy lipid partitioning and membrane translocation mainly reflecting passive transmembrane distribution in accordance with the imposed transmembrane potential. Passive membrane translocation is currently the most supported mechanism [13,14]. Based on this mechanism, the uptake of sestamibi is generally associated with the presence of several mitochondria with a high membrane potential [15].

Despite the important chemical-physical and clinical evidence regarding the use of BSGI with [99mTc]Tc-Sestamibi in the management of breast cancer patients reported above, few studies have been performed on the cellular/molecular modifications of breast cancer tissues induced by the sestamibi uptake. In fact, only indirect data about the association between [99mTc]Tc-Sestamibi uptake and the cellular/molecular characteristics of breast cancer cells, such as their proliferation index, have been investigated. In this context, Erba et al. reported very preliminary data about the role of [99mTc]Tc-Sestamibi uptake as an indicator of chemotherapy induced apoptosis [16]. However, no definitive clinical or in vitro experimental data are currently available.

Starting from these considerations, this study aims to investigate the possible role of [99mTc]Tc-Sestamibi in the regulation of biological processes involved in cancer progression such as proliferation and apoptosis ex vivo and in vitro. To this end, the in vivo values of [99mTc]Tc-Sestamibi uptake have been associated with the in situ expression of both Ki67 (proliferation index) and caspase-3 (apoptosis). In addition, in vitro investigations using a breast cancer cell line (BT474) have been performed to study the possible cellular and molecular modifications of cancer cells following [99mTc]Tc-Sestamibi uptake.

## 2. Materials and Methods

The Policlinico Tor Vergata ethical committee approved this protocol with the reference number # 129.18, 26 July 2018. Furthermore, all methodologies and experimental procedures described herein were achieved in agreement with the last Helsinki Declaration.

Exclusion criteria were a second cancer and neoadjuvant hormonal or radiation therapy prior to surgery.

According to these criteria, we retrospectively enrolled 40 consecutive breast cancer patients (58.36 $\pm$ 1.99 years; range 42–65 years) who underwent both BSGI with [99mTc]Tc-Sestamibi and breast biopsy procedures.

For each of them, histological diagnosis and immunohistochemical investigations were performed.

### 2.1. [99mTc]Tc-Sestamibi-High Resolution SPECT

BSGI with [99mTc]Tc-Sestamibi investigations were performed as described in a previous study [17]. Briefly, a BSGI scan was performed in 10–15 min following an intravenous administration of 740 MBq [99mTc]Tc-Sestamibi through an antecubital vein contralateral

to the suspicious breast side to avoid potential false-positive uptake in the axillary lymph nodes. The patients remained seated during the procedure. Craniocaudal and mediolateral oblique (MLO) images were obtained in both breasts using high-resolution BSGI.

All 40 patients had biopsy. BSGI was performed before biopsy in 25 patients and after biopsy in 15 patients. When BSGI was performed after biopsy, the minimum interval between biopsy and imaging was 7 days in an attempt to avoid the effects of post-biopsy inflammation as much as possible.

For the qualitative analysis of the BSGI, two investigators classified positive and negative findings. Lesions with no demonstrable uptake and those with diffuse heterogeneous or minimal patchy uptake were considered negative, whereas lesions with scattered patchy uptake, partially focal uptake, or any other focal uptake were regarded as positive. Irregular-shaped regions of interest (ROIs) were used to encase the lesions. The evaluation of the lesion to nonlesion ratio (L/N) was estimated according to the study of Tan et al. [18]. For patients who underwent BSGI with [99mTc]Tc-Sestamibi before biopsy (n = 25), the BSGI-guided biopsy procedure was performed as previously described [19].

### 2.2. Histology

Breast biopsy samples were formalin fixed and embedded in paraffin. Serial sections were used for both hematoxylin-eosin (H&E) and immunohistochemistry staining [20].

### 2.3. Immunohistochemistry

Immunohistochemistry was used to study the expression of an apoptotic in situ biomarker, caspase-3, and the proliferation index by Ki67. Paraffin sections with a thickness of 3 μm were treated with EDTA citrate buffers at a pH of 7.8 for 30 min at 95 °C to antigen retrieval reaction. Afterwards, sections were incubated with the primary antibodies diluted 1:100 for 60 min at room temperature; anti-Ki67 rabbit monoclonal antibody (clone 30-9, Ventana, Tucson, AZ, USA) and anti-caspase-3 mouse monoclonal antibody (31A1067, Novus Biologicals, Centennial, CO, USA). Washing was performed with PBS/Tween 20 with a pH of 7.6 (UCS Diagnostic, Morlupo, RM, Italy).

Digital scanning was used to evaluate the immunohistochemical reactions (Iscan Coreo, Ventana, Tucson, AZ, USA). Specifically, digital images from caspase-3 reactions were evaluated in a semi-quantitative approach by counting the number of positive breast cancer cells (out of a total of 500 in randomly selected regions). Ki67 was calculated in terms of percentage of positive cancer cells. Reactions have been set up by using specific positive and negative control tissues.

### 2.4. Cell Culture

BT-474 cells obtained from the American Type Culture Collection (ATCC. Manassas, VA, USA) and maintained by the Cell and Tissue Culture Core, Lombardi Cancer Center (Reservoir Rd. NW Washington D.C. 20057, USA). Cells were routinely cultured in DMEM high glucose (Sigma-Aldrich, St. Louis, MO, USA) supplemented with 10% fetal bovine serum (FBS).

In detail, cells from the first or second passage were seeded into a 24-well plate at a density of $30 \times 10^3$ cells/well. Successively, BT-474 cells were incubated with: (a) [99mTc]Tc-Sestamibi 10 μg/mL, (b) [99mTc]Tc-Sestamibi 1 μg/mL, and (c) [99mTc]Tc-Sestamibi 0.1 μg/mL. The expression of both Ki67 and caspase-3 were evaluated at T0 and after 24, 48, 72, and 120 h after sestamibi incubation. Cells treated with the vehicle (lyophilisate resuspended) were used as a control (CTRL).

Cell proliferation was investigated both by counting the number of cells for each time point, and by bromodeoxyuridine incorporation assay performed at time 0 and after 72 h. Morphology was studied by both toluidine blue staining.

*2.5. Immunocytochemistry*

Immunocytochemistry was performed to investigate the expression of caspase-3 and Ki67 on BT-474 cells treated with 99mTc-Sestamibi. Caspase-3 was evaluated by immunoperoxidase analysis, while Ki67 expression was evaluated by immunofluorescence staining in order to reduce the background.

BT-474 Cells were plated on poly-l-lysine coated slides (Sigma-Aldrich cat #P4707) in 24-well cell culture plates and fixed in 4% paraformaldehyde. After pre-treatment with EDTA citrate at 95 °C for 20 min and 0.1% Triton X-100 for 15 min, cells were incubated for 1 h with the anti-Ki67 rabbit monoclonal antibody (clone 30-9, Ventana, Tucson, AZ, USA) and anti-caspase-3 mouse monoclonal antibody (31A1067, Novus Biologicals, Littleton, CO, USA). Washing was performed with PBS/Tween20 with a pH of 7.6. Regarding the study of caspase-3 expression, reactions were revealed by using an horseradish peroxidase-3,3′-Diaminobenzidine detection kit (UCS Diagnostic, Rome, Italy). Conversely, the Ki67 signal was revealed by using a TexasRed conjugate anti-rabbit antibody.

Reactions were evaluated by counting the number of Ki67 or caspase-3-positive cells of 500 in total in randomly selected regions.

*2.6. TEM and EDX Analysis of Cell Cultures*

Cells were fixed in 4% paraformaldehyde, post-fixed in 2% osmium tetroxide and embedded in EPON resin for morphological studies. After washing with 0.1 M phosphate buffer, the sample was dehydrated by a series of incubations in 30%, 50%, and 70% ethanol. Dehydration was continued by incubation steps in 95% ethanol, absolute ethanol, and hydroxypropyl methacrylate, then samples were embedded in EPON (Agar Scientific, Stansted, Essex, UK).

Eighty μm ultra-thin sections were mounted onto copper grids and observed with a Hitachi 7100FA transmission electron microscope (Hitachi, Schaumburg, IL, USA) to study the mitochondria ultrastructure.

Unstained ultra-thin sections with a thickness of approximately 100 nm were mounted onto copper grids for microanalysis. EDX spectra were acquired with a Hitachi 7100FA transmission electron microscope (Hitachi, Schaumburg, IL, USA) and an EDX detector (Thermo Scientific, Waltham, MA, USA).

*2.7. Statistical Analysis*

Statistical analysis was performed using GraphPad Prism 5 software (San Diego, CA, USA). Immunohistochemical data were analyzed by the Kruskal-Wallis test ($p < 0.05$) and by Mann-Whitney test ($p < 0.0005$). In vitro data about the number of Ki67-positive cells and caspase-3-positive cells were analyzed by using one-way ANOVA ($p < 0.05$).

**3. Results**

*3.1. 99mTc-Sestamibi-High Resolution SPECT Analysis*

BSGI with [99mTc]Tc-Sestamibi showed the uptake of radiopharmaceuticals in all 35 breast cancer patients (L/N max 5.09; min 1.43) (Figure 1A). No significant differences concerning L/N Ratio were observed among breast cancer histotypes.

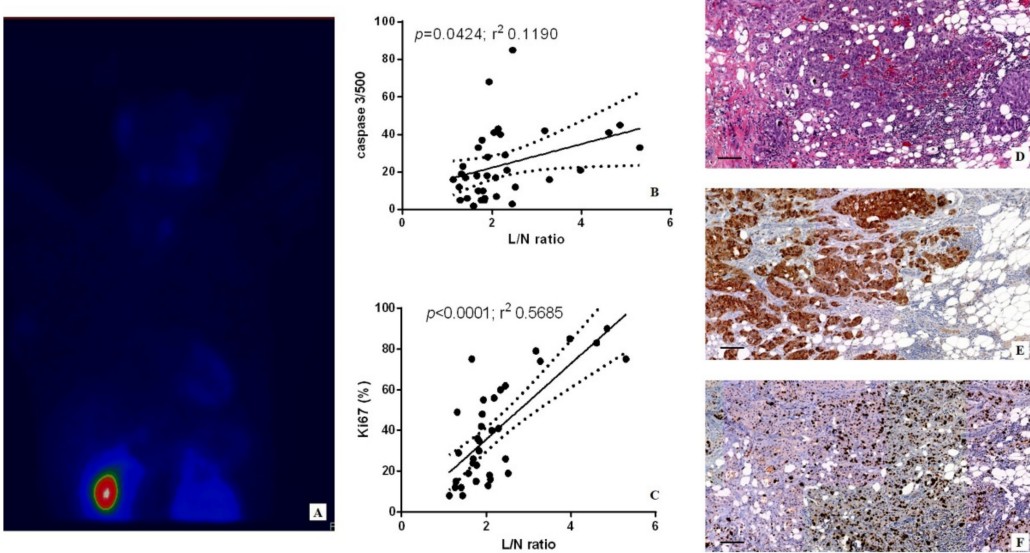

**Figure 1.** Breast-specific γ imaging with [99mTc]Tc-Sestamibi and both caspase-3 and Ki67 expression. (**A**) Maximum-intensity projection of breast-specific γ imaging with [99mTc]Tc-Sestamibi in a 69-year-old breast cancer patient. (**B**) The graph shows the positive and significant association between the number of caspase-3-positive breast cancer cells and the sestamibi uptake (L/M ratio). (**C**) The graph displays significant association between the number of Ki67-positive breast cancer cells and the sestamibi uptake (L/M ratio). (**D**) T hematoxylin and eosin section shows G3 infiltrating breast carcinomas (scale bar represents 50 μm). (**E**) The image displays numerous caspase-3-positive breast cancer cells (scale bar represents 50 μm). (**F**) High percentage of Ki67-positive breast cancer cells (scale bar represents 50 μm).

### 3.2. Histology

Breast biopsies were classified according to the Nottingham histological system [21]. In particular, 8/40 G1 infiltrating carcinomas, 14/40 G2 infiltrating carcinomas, and 9/40 G3 infiltrating carcinomas were found.

### 3.3. [99mTc]Tc-Sestamibi Uptake vs. Apoptosis

To investigate the possible association between sestamibi uptake and the apoptotic phenomenon, linear regression analyses have been performed by comparing the L/N ratio values and the number of caspase-3-positive cells. Linear regression analysis showed a positive association between sestamibi uptake and the number of caspase-3-positive breast cancer cells ($p$ = 0.0424; $r^2$ 0.1190) (Figure 1B,D,E), even though the absolute number of apoptotic cells were low, especially if compared with the number of Ki67-positive cells (Figure 1C,D).

### 3.4. [99mTc]Tc-Sestamibi Uptake vs. Proliferation Index

The percentages of Ki67-positive breast cancer cells, evaluated by immunohistochemistry, have been used as proliferation index values. According to our previous investigation, a positive and significant correlation between sestamibi uptake and the proliferation index was observed ($p$ < 0.0001; $r^2$ 0.5685) (Figure 1C,F).

### 3.5. Effect of Sestamibi on Breast Cancer Cells: In Vitro Study

At the end of each experimental point, BT-474 cells were fixed with 4% paraformaldehyde and used to evaluate the number of both Ki67- and caspase-3-positive cells. One-way ANOVA showed significant data distribution for both the number of caspase-3- ($p$ < 0.0001) and Ki67- ($p$ = 0.0038) positive cells. Notably, a high concentration of [99mTc]Tc-Sestamibi (10 μg/mL) induced a significant increase in the number of apoptotic cells (caspase-3-positive cells) when compared with all other experimental conditions, including the control

(Figure 2A–I). Specifically, a great increase in the number of caspase-3-positive cells was observed after 48h (Figure 2A,C). This datum suggests that [99mTc]Tc-Sestamibi could be able to induce the apoptotic process by caspase-3 signal only at high concentrations. In other experimental conditions, no significant effects were detected (Figure 2A).

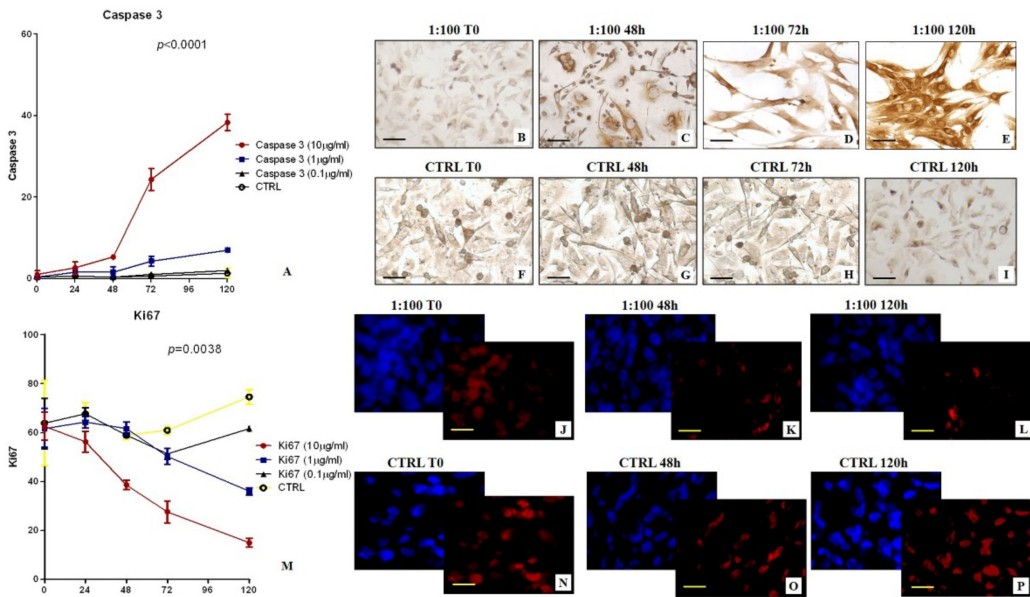

**Figure 2.** In vitro evaluation of the effect of [99mTc]Tc-Sestamibi on BT474 breast cancer cells. (**A**) The graph shows the number of caspase-3-positive BT474 breast cancer cells after sestamibi treatment. (**B**) No/rare caspase-3-positive cells at T0. (**C**) The image shows caspase-3-positive cells after 48 h of [99mTc]Tc-Sestamibi incubation (10 µg/mL). (**D**) Numerous caspase-3-positive cells after 72 h of [99mTc]Tc-Sestamibi incubation (10 µg/mL). (**E**) High number of caspase-3-positive cells after 120 h of [99mTc]Tc-Sestamibi incubation (10 µg/mL). (**F–I**) No/rare caspase-3-positive cells at each time point in the absence of [99mTc]Tc-Sestamibi incubation. Scale bar represents 50µm for (**B–I**) images. (**J**) Immunofluorescence displays numerous Ki67-positive breast cancer cells (blue DAPI-Texas red Ki67) at T0. (**K**) Some Ki67-positive breast cancer cells (blue DAPI-Texas red Ki67) after 48 h of [99mTc]Tc-Sestamibi incubation (10 µg/mL). (**L**) Rare Ki67-positive breast cancer cells (blue DAPI-Texas red Ki67) after 120 h of 10 µg/mL [99mTc]Tc-Sestamibi incubation. (**M**) The graph shows the number of caspase-3-positive BT474 breast cancer cells after sestamibi treatment. (**N–P**) Numerous Ki67-positive breast cancer cells (blue DAPI-Texas red Ki67) at each time point in the absence of [99mTc]Tc-Sestamibi incubation (10 µg/mL). Scale bar represents 20 µm for (**J–L,N–P**) images.

Concerning the cell proliferation (Ki67), a significant decrease in the number of Ki67-positive cells was observed in cell cultures incubated with both 10 µg/mL and 1 µg/mL of [99mTc]Tc-Sestamibi when compared with cells treated with a concentration of 1 µg/mL (Figure 2M–P). Indeed, the data reported here showed that a low concentration of [99mTc]Tc-Sestamibi (1 µg/mL) does not influence the proliferation index (Figure 2M). Cell cultures treated with both 1 µg/mL and 0.1 µg/mL [99mTc]Tc-Sestamibi displayed a proliferation index similar to that observed in the control (Figure 2M). This datum can explain the association observed in vivo between sestamibi uptake and the percentage of Ki67-positive cells. It is important to note that high concentration of [99mTc]Tc-Sestamibi (10 µg/mL) induced a significant reduction in the proliferation index already after 24 h of treatment (Figure 2M–L).

By comparing the value of Ki67 and caspase-3 at every time point it became clear that by only using 10 µg/mL of [99mTc]Tc-Sestamibi concentration, a complete reversion between proliferation and apoptosis was obtained (Figure 3A–C). In particular, after 72h, the number of caspase-3-positive cells was higher than those positive for Ki67, suggesting an imbalance capable of arresting the tumor's proliferation (Figure 3A). Morphological

evaluation confirmed the presence of apoptotic bodies in the cell cultures cultured with a high concentration of [99mTc]Tc-Sestamibi (10 µg/mL) (Figure 3D–G). Transmission electron microscopy investigations showed numerous apoptotic bodies in cell cultures incubated with 10 µg/mL [99mTc]Tc-Sestamibi (Figure 3E–G). The damages were evident after 72 h. EDX microanalysis frequently showed the presence of [99mTc]Technetium in the cytoplasm of apoptotic cells (Figure 3F,G) or in the mitochondria of the cell in the early phases of apoptotic process.

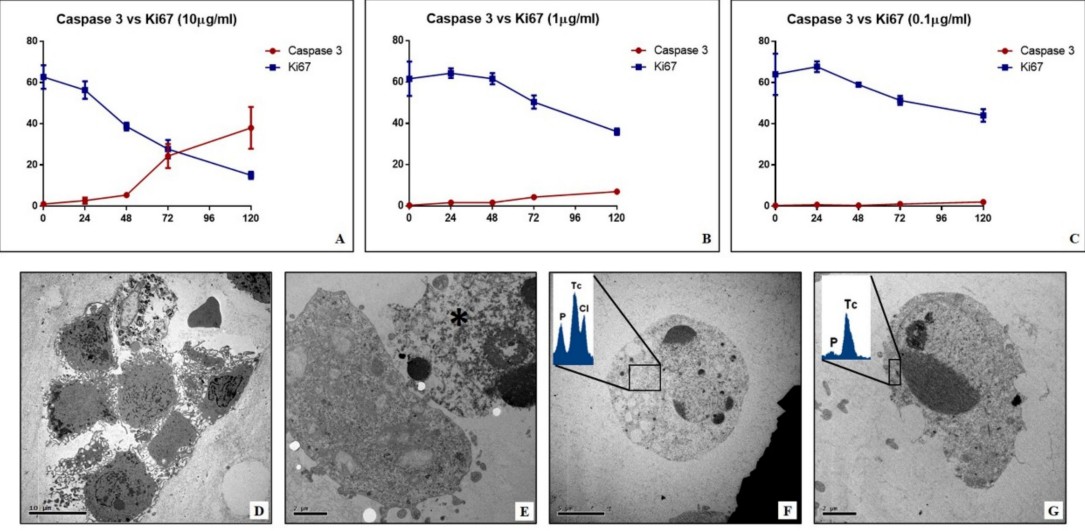

**Figure 3.** Comparison between apoptosis and proliferation of BT474 breast cancer cells treated with [99mTc]Tc-Sestamibi and ultrastructural analysis. (**A**) Graph shows the number of both caspase-3- and Ki67-positive BT474 breast cancer cells after 24, 48, 72, 96, and 120 h of [99mTc]Tc-Sestamibi incubation (10 µg/mL). (**B**) Graph displays the number of both caspase-3- and Ki67-positive BT474 breast cancer cells after 24, 48, 72, 96, and 120 h of [99mTc]Tc-Sestamibi incubation (10 µg/mL). (**C**) Graph shows the number of both caspase-3- and Ki67-positive BT474 breast cancer cells after 24, 48, 72, 96, and 120 h of [99mTc]Tc-Sestamibi incubation (10 µg/mL). (**D**) Electron micrograph displays ultrastructure of BT474 breast cancer cells (CTRL T0). (**E**) Image shows an apoptotic breast cancer cell (asterisk) close to a non-apoptotic cell (10 µg/mL at 48 h). (**F,G**) EDX spectrum revealed the presence of [99mTc]Technetium into the cytoplasm of apoptotic cells (10 µg/mL at 120 h).

## 4. Discussion

BSGI with [99mTc]Tc-Sestamibi can represent a remarkable opportunity to detect both primary and metastatic lesions characterized by the presence of breast cancer cells with high amounts of mitochondria. In fact, the current data about the chemical-physical characteristics of sestamibi, as well as its pharmacokinetics, seem to indicate its tropism for the accumulation in the cell organelles with a negative membrane potential, such as mitochondria. In support of this, it is known that once in the cytoplasm, sestamibi may translocate into the mitochondria according to its cationic nature [15,16]. The positive charge on sestamibi may drive this molecule into the mitochondria during cell metabolic activities that increase the negative plasma membrane potential. Despite this evidence, few studies have been performed about the possible effect of sestamibi uptake in breast cancer cells. The accumulation of sestamibi in the cell allows it to hypothesize the alteration of normal mitochondria functions and consequently of the cancer cell's homeostasis, mainly at the level of cell proliferation and cell death.

Ex vivo data reported here strengthened the known association between sestamibi uptake and Ki67 expression, thus confirming the capability of BSGI with [99mTc]Tc-Sestamibi to identify breast cancer lesions characterized by high proliferation index [17]. In line with this, Urbano et al. also demonstrated the association between sestamibi uptake and proliferation index in parathyroid lesions [18]. Surprisingly, linear regression analysis also

showed a significant association between the sestamibi uptake and the number of apoptotic cells evaluated as caspase-3-positive cells. Caspase-3 is a protein effector involved in apoptosis [19]. Activated caspase-3, as well as caspase-6 and caspase-7, may cleave multiple structural and regulatory proteins that are critical for cell survival and maintenance [18]. However, caspase-3 is considered the most important executioner caspase and it is involved in both intrinsic and extrinsic apoptotic pathways [20]. Alteration in both mitochondrial function and structure induce the release of mediators such as BCL-2 family proteins that amplify caspase-3 activity, inducing the intrinsic pathway of apoptosis [21]. Thus, the expression of caspase-3 in breast cells can be related to the occurrence of intrinsic apoptosis. Data on both Ki67 and caspase-3, despite being in disagreement, can be related to the same processes involved in breast cancer progression. Indeed, as demonstrated by Ryoo and Bergmann [22], the caspase-3 related apoptotic phenomenon may be related to cancer progression due to the communication between apoptotic cells and surrounding ones. This biological process occurs physiologically during embryogenesis, where proapoptotic proteins—mostly caspases—can induce the proliferation of neighboring surviving cells to replace dying cells. It is demonstrated that the deregulation of this process in cancer tissues could sustain tumor proliferation and progress. Thus, apparently only our in vivo data seem conflicting. The association of sestamibi uptake with both proliferation index and apoptosis could be considered an instrumental/clinical manifestation of pathophysiological mechanisms commonly involved in breast cancer development. However, a molecular characterization of the events associated with the accumulation of sestamibi into the mitochondria could open new clinical perspectives in the management of breast cancer cells. In fact, the regulation of the mitochondria related apoptotic process is currently one of the main targets of anti-cancer therapies.

In this context, the preliminary data of Erba and colleagues suggested that the sestamibi uptake correlated with apoptosis levels in breast cancer tissues following chemotherapy treatment [23].

In this study, the biological link between sestamibi uptake and the apoptotic phenomenon has been investigated in vitro by using BT-474 breast cancer cell lines. In particular, three different concentrations of [99mTc]Tc-Sestamibi (10 μg/mL, 1 μg/mL, 0.1 μg/mL) have been used to investigate the response of breast cancer cells to sestamibi uptake. Concerning the proliferation index, a significant decrease was observed in breast cancer cells incubated with a high concentration of [99mTc]Tc-Sestamibi (10 μg/mL). It is important to note that both in the control and at a lower concentration of [99mTc]Tc-Sestamibi (0.1 μg/mL), no decrease in proliferation index was observed after 120h. Excluding the experiment with a high concentration of [99mTc]Tc-Sestamibi (10 μg/mL), in vitro data confirmed the uptake of sestamibi in breast cancer cells during the proliferation phase. Amazingly, in vitro investigations showed a significant increase in caspase-3-positive cells in cultures incubated with 10 μg/mL of 99mTc-sestamibi. In particular, after 72 h, the presence of sestamibi in the cell medium induced a complete inversion between the proliferation index and apoptosis. In fact, at this time point, the number of caspase-3-positive cells was greater than those positive to the Ki67 markers. From a morphological point of view, these cell cultures showed numerous dead cells characterized by typical apoptotic sign such as pyknosis, or karyopyknosis. In addition, transmission electron microscopy investigations allowed us to evaluate the ultrastructure of breast cancer cells incubated with [99mTc]Tc-Sestamibi, demonstrating a progressive increase in the presence of apoptotic alterations. These damages were associated with both [99mTc]Tc-Sestamibi concentration and incubation time. Remarkably, EDX investigations also showed the presence of [99mTc]Technetium in the cytoplasm of apoptotic cells.

Altogether, the data of this study suggest that, in the early phases, sestamibi uptake occurs mainly in breast cancer cells with a high proliferation index because of the high metabolic cell activity and the subsequent increase in the mitochondrial membrane potential. Then, the accumulation of sestamibi in the breast cancer cell mitochondria can induce an alteration of normal homeostasis, thus triggering the apoptotic event.

## 5. Conclusions

For the first time, this study reported ex vivo and in vitro data about the correlation between sestamibi uptake and apoptosis, suggesting the possible role of sestamibi in the regulation of the pathophysiological processes involved in breast cancer. The evidence of the accumulation of sestamibi in breast cancer cells and the subsequent mitochondrial damage can open new clinical perspectives on the use of this radiopharmaceutical in both the diagnosis and treatment of breast cancers. If confirmed by further ex vivo and in vitro studies, the capability of sestamibi to induce apoptosis of breast cancer cells can lay down the scientific rationale for considering this molecule as a theragnostic agent. Indeed, different concentrations of [99mTc]Tc-Sestamibi could be used for the detection of cancer lesions with a high proliferation index or to stimulate apoptosis, thus countering cancer growth. Lastly, these investigations further highlight the fundamental cooperation between nuclear medicine and general diagnostic imaging, and the pathology in both research and diagnostic applications [24,25].

**Author Contributions:** Conceptualization: N.U., M.S., and O.S.; methodology: M.S., R.B., E.B., and A.M.; formal analysis R.B., and E.B.; data curation: A.M., N.U. and O.S.; writing—original draft preparation: N.U., M.S., and O.S.; writing—review and editing: R.B., E.B. and A.M.; supervision, O.S. All authors have read and agreed to the published version of the manuscript.

**Funding:** This research received no external funding.

**Institutional Review Board Statement:** The "Policlinico Tor Vergata" ethical committee approved this protocol with the reference number no. 129.18, 26 July 2018. Furthermore, all methodologies and experimental procedures herein described were achieved in agreement with the last Helsinki Declaration.

**Informed Consent Statement:** Informed consent was obtained from all subjects involved in the study.

**Data Availability Statement:** The data presented in this study are available on request from the corresponding author. The data are not publicly available due to privacy concerns.

**Conflicts of Interest:** The authors declare no conflict of interest.

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
