# Peer review of "[99mTc]Tc-Sestamibi Bioaccumulation Can Induce Apoptosis in Breast Cancer Cells: Molecular and Clinical Perspectives"

_applsci, doi:10.3390/app11062733_

Round 1

Reviewer 1 Report

This manuscript presents the results of a pilot study. the aim was to investigate the possible role of 99mTc-sestamibi in the processes of proliferation and apoptosis of cancer cells (breast cancer). The authors of the study, based on the analysis of a small number of patients (40) and in vitro investigations on BT-474 cells, suggested the possible role of sestamibi in the regulation of the pathophysiological process involved in breast cancer. If confirmed by further ex vivo and in vitro studies, the ability of 99mTc-sestamibi to induce apoptosis of breast cancer cells may constitute the scientific rationale for considering this molecule as a teragnostic agent.

This pilot study is interesting but a few minor points need attention:

  1. Attention to the correct writing of the name of the radiopharmaceutical in the various points of the manuscript which must be the following: [99mTc]Tc-Sestamibi and not: 99TC-sestamibi or 99mTC sestamibi or sestamibi;
  2. Attention to the correct writing of the name of the technetium (line256): [99mTc]Technetium
  3. Line 139: Cells treated with the vehicle were used as control 139 (CTRL). Specify the composition. Is it the lyophilisate resuspended with saline alone?

I would recommend the acceptance of the manuscript after appropriate modifications. 

Author Response

This manuscript presents the results of a pilot study. the aim was to investigate the possible role of 99mTc-sestamibi in the processes of proliferation and apoptosis of cancer cells (breast cancer). The authors of the study, based on the analysis of a small number of patients (40) and in vitro investigations on BT-474 cells, suggested the possible role of sestamibi in the regulation of the pathophysiological process involved in breast cancer. If confirmed by further ex vivo and in vitro studies, the ability of 99mTc-sestamibi to induce apoptosis of breast cancer cells may constitute the scientific rationale for considering this molecule as a teragnostic agent.

Reply: we would like to thank the Reviewer for expressing interest in our work, and for their availability to review a revised version of our manuscript. In the revised form of our manuscript, we included all changes suggested by the reviewer.

  1. Attention to the correct writing of the name of the radiopharmaceutical in the various points of the manuscript which must be the following: [99mTc]Tc-Sestamibi and not: 99TC-sestamibi or 99mTC sestamibi or sestamibi

Reply: thanks for this point out. We corrected the manuscript according to the editor suggestion.

  1. Attention to the correct writing of the name of the technetium (line256): [99mTc]Technetium

Reply: we corrected this in all text.

  1. Line 139: Cells treated with the vehicle were used as control 139 (CTRL). Specify the composition. Is it the lyophilisate resuspended with saline alone?

Reply: we specified the nature of the control solution in the text (lyophilisate resuspended).

Reviewer 2 Report

Corrections to be done:

  • Line 56: also be used for early detection (instead of: to)
  • Line 61: translocation mainly reflecting passive (instead of: associated to
  • Line 64: associated with the presence (instead of: to)
  • Line 64: with a high membrane (instead of: an)
  • Line 72: Sestamibi uptake as an indicator of the chemotherapy (instead of: indicators
  • Line 75-76: this study aims to investigate the possible role 99mTC-sestamibi in the regulation of biological process involved in cancer progression such as proliferation and apoptosis ex vivo and in vitro 
  • Line 136: 30 x 103 (instead of : 30 x103)
  • Line 194: especially (instead of: execially)
  • Line 198: previous investigation (instead of: previusly)
  • Line 209: Caspase 3 positive breast cancer cells (instead of: caspase 5)
  • Line 224: a concentration (instead of: an)
  • Line 270: can represent (instead of : represents)
  • Line 272: the current data (instead of : currently)
  • Line 277:may drive (instead of: drives)
  • Line 280: hypothesize (instead of: hypothesis)
  • Line 283/284: remove the first sentence because it is a repetition 
  • Line 302: intrinsic pathway (instead of: way)
  • Line 337: dead cells (instead of: death
  • Line 354: diagnosis and treatment (instead of: cure)
  • Line 360: in general diagnostic imaging (instead of :imaging diagnostic) 

Comment: In the 3rd paragraph of the introduction: during listing the 3 models the authors listed just 2 models a) and b). The 3rd model should be mentioned 

Author Response

Line 56: also be used for early detection (instead of: to)

Line 61: translocation mainly reflecting passive (instead of: associated to)

Line 64: associated with the presence (instead of: to)

Line 64: with a high membrane (instead of: an)

Line 72: Sestamibi uptake as an indicator of the chemotherapy (instead of: indicators)

Line 75-76: this study aims to investigate the possible role 99mTC-sestamibi in the regulation of biological process involved in cancer progression such as proliferation and apoptosis ex vivo and in vitro

Line 136: 30 x 103 (instead of : 30 x103)

Line 194: especially (instead of: execially)

Line 198: previous investigation (instead of: previusly)

Line 209: Caspase 3 positive breast cancer cells (instead of: caspase 5)

Line 224: a concentration (instead of: an)

Line 270: can represent (instead of : represents)

Line 272: the current data (instead of : currently)

Line 277:may drive (instead of: drives)

Line 280: hypothesize (instead of: hypothesis)

Line 283/284: remove the first sentence because it is a repetition

Line 302: intrinsic pathway (instead of: way)

Line 337: dead cells (instead of: death)

Line 354: diagnosis and treatment (instead of: cure)

Line 360: in general diagnostic imaging (instead of :imaging diagnostic)

Comment: In the 3rd paragraph of the introduction: during listing the 3 models the authors listed just 2 models a) and b). The 3rd model should be mentioned

Reply: we would like to thank the Reviewer for expressing interest in our work, and for their availability to review a revised version of our manuscript.

We performed all corrections suggested by the reviewer.

Round 2

Reviewer 1 Report

You have successfully used my little tips.
My opinion on your paper is now fully positive